# Potentially Pathogenic *SORL1* Mutations Observed in Autosomal-Dominant Cases of Alzheimer’s Disease Do Not Modulate APP Physiopathological Processing

**DOI:** 10.3390/cells12242802

**Published:** 2023-12-08

**Authors:** Charlotte Bauer, Eric Duplan, Peter Saint-George-Hyslop, Frédéric Checler

**Affiliations:** 1INSERM, CNRS, IPMC, Team Labeled “Laboratory of Excellence (LABEX) Distalz”, Université Côte d’Azur, 660 Route des Lucioles, Sophia-Antipolis, 06560 Valbonne, France; bauer@ipmc.cnrs.fr (C.B.); duplan@ipmc.cnrs.fr (E.D.); 2Center for Research in Neurodegenerative Diseases, Department of Medicine, Toronto Western Hospital Research Institute, University Health Network, University of Toronto, Toronto, ON M5G 1L7, Canada; p.hyslop@utoronto.ca

**Keywords:** SORL1, mutations, secretases, APP, Aβ peptides, C-terminal fragments, neprilysin, proteasome, degradation, cellular localization, transient and stable expressions

## Abstract

The *SORL1* gene encodes LR11/SorLA, a protein that binds β-amyloid precursor protein (APP) and drives its intracellular trafficking. *SORL1* mutations, occurring frequently in a subset of familial cases of Alzheimer’s disease (AD), have been documented, but their pathogenic potential is not yet clear and questions remain concerning their putative influence on the physiopathological processing of APP. We have assessed the influence of two *SORL1* mutations that were described as likely disease-causing and that were associated with either benign (SorLA^924^) or severe (SorLA^511^) AD phenotypes. We examined the influence of wild-type and mutants SorLA in transiently transfected HEK293 cells expressing either wild-type or Swedish mutated APP on APP expression, secreted Aβ and sAPPα levels, intracellular Aβ 40 and Aβ42 peptides, APP-CTFs (C99 and C83) expressions, α-, β- and γ-secretases expressions and activities as well as Aβ and CTFs-degrading enzymes. These paradigms were studied in control conditions or after pharmacological proteasomal modulation. We also established stably transfected CHO cells expressing wild-type SorLA and established the colocalization of APP and either wild-type or mutant SorLA. SorLA mutations partially disrupt co-localization of wild-type sorLA with APP. Overall, although we mostly confirmed previous data concerning the influence of wild-type SorLA on APP processing, we were unable to evidence significant alterations triggered by our set of SorLA mutants, whatever the cells or pharmacological conditions examined. Our study , however, does not rule out the possibility that other AD-linked *SORL1* mutations could indeed affect APP processing, and that pathogenic mutations examined in the present study could interfere with other cellular pathways/triggers in AD.

## 1. Introduction

Alzheimer’s disease (AD) is a complex pathology characterized by either early or late onsets. Most of early onset (EOAD) cases are of familial origin and follow an autosomal dominant transmission [1,2,3], while late onset (LOAD) cases usually referred to as sporadic AD cases have been shown to be associated with various risk factors [4,5]. The recent occurrence of genome wide association studies allowed elucidating some of these risk factors [6,7]. Indeed, cohorts built with a growing number of cases coupled to powerful bioinformatics allowed analysis of this bulk data and led to a still increasing number of gene candidates that could underlie individual susceptibility to AD pathology.

Contradicting the apparent dichotomy between familial and sporadic AD cases, *SORL1* gene has been implicated in both EOAD and LOAD. Thus, Rogaeva and Colleagues [8] first evidenced a genetic link between *SORL1* and AD. This was further corroborated by independent meta-analyses confirming the link between *SORL1* variants and sporadic AD cases [9,10,11]. Interestingly, exome sequencing also delineated *SORL1* mutations in both EOAD and LOAD cases [12,13].

*SORL1* encodes a sortilin-related receptor with A-type repeats named SorLA (also referred to as LR11) [8]. This protein has been characterized as a neuronal sorting receptor for β-amyloid precursor protein (APP), the precursor of Aβ peptides that accumulate in AD-affected brains [14]. The ability of SorLA to physically interact with APP drives its lysosomal sorting and was shown to be affected by familial AD mutations [13,15]. This was reported to yield functional consequences on the levels of Aβ load in cells [13,16] and animals [15]. However, it appears that the phenotypic alterations on Aβ peptides levels as well as the nature of Aβ peptides modified varied according to the degree of rarity of the *SORL1* variants [16] as well as cellular and experimental settings. Indeed, unlike in the present work, no studies exhaustively examined the direct influence of SorLA mutations on other steps of the APP physiopathological maturation.

Here, we exhaustively examined the influence of two frequent *SORL1* mutations linked to either benign of severe autosomal dominant cases of AD on the amyloidogenic and non-amyloidogenic APP products, secretases expressions and activity and Aβ degradation processes.

## 2. Materials and Methods

### 2.1. Constructs and Site-Directed Mutagenesis

Wild-type human *SORL1* cDNA, cloned in the pcDNA3.1 (+) vector, was provided by one of us (P. St. G-H). Site-directed mutagenesis kits from Agilent (Santa Clara, CA, USA, QuikChange II) and mutagenesis primers (Eurogentec, Seraing, Belgium) were used to obtain the SorLA^511^ and SorLA^924^ mutants (Table 1). All nucleotides modifications were verified by full sequencing of the constructs.

### 2.2. Cell Cultures and Transfections

Human Embryonic Kidney 293 cells (HEK293) expressing wild-type APP (wt-APP) or swedish-mutated APP (swe-APP) were cultured (5% CO_2_, 37 °C) in DMEM (Invitrogen, Carlsbad, CA, USA) containing Fetal Bovine Serum (FBS 10%, Sigma, St. Louis, MO, USA) and antibiotics (50 U/mL penicillin/50 μg/mL streptomycin, Invitrogen). Wild-type or mutant *SORL1* cDNA were transiently transfected (24 h) using the jetPRIME reagent (Polyplus, Strasbourg, France) (2 μg or 10 μg of cDNA in 35 mm or 100 mm dishes, respectively) in control or inhibitor conditions (lactacystin, 5 μM). In alkalizing conditions, cells were treated for 16 h at 37 °C with NH4Cl (10 mM). Chinese Hamster Ovary cells (CHO) expressing wt-APP were obtained by stable transfection of the pcDNA4 vector [17]. Cells were maintained in DMEM containing 10% FBS, sodium hypoxanthine-thymidine supplement, and 300 μM proline. Cells were stably transfected with 2 μg of wild-type or mutants SORL1 cDNA according to Lipofectamine protocols (Thermo Fisher Scientific, Waltham, MA, USA). Clones were selected with 250 μg/mL Zeocin (Invitrogen, Carlsbad, CA, USA).

### 2.3. Cells Immunostaining

CHO cells grown on coverslips were fixed in paraformaldehyde 4% solution for 10 min, permeabilized with Triton-X 100 (0.1%) for 10 min, saturated in BSA (5%)/Tween20 (0.1%), and probed for 1 h with appropriate primary antibodies: β-Amyloid (clone 6E10, Sigma, St. Louis, MO, USA) mouse monoclonal (1:1000) or SorLA rabbit polyclonal (1:1000, Sigma, St. Louis, MO, USA). After washes, coverslips were incubated for 1 h with Alexa Fluor-488 and Alexa Fluor-594 conjugated antibodies (Molecular Probes, Eugene, OR, USA, 1:1000) and DAPI (1:20,000, Roche, Basel, Switzerland) staining. Finally, the sections were washed with PBS, then mounted onto glass slides and cover-slipped. The stained slices were kept at 4 °C before analysis with confocal microscopy (Zeiss LSM 780 with 63X Objective).

### 2.4. sAPPα Secretion Andimmunoprecipitation of Total Secreted Aβ

Cells were grown in 6-well plates and allowed to secrete Aβ for 16 h in OptiMEM (1 mL, Invitrogen, Carlsbad, CA, USA) + 1% FBS (Sigma, St. Louis, MO, USA) containing phosphoramidon (10 μM, Sigma, St. Louis, MO, USA) in order to prevent Aβ degradation by neprilysin as described [18]. Media were collected, completed with one tenth of 10X RIPA buffer (Tris-HCl pH 8.0 (100 mM) containing NaCl (1.5 M) and EDTA (50 mM)). sAPPα secretion was measured in 20 μL of medium collected and deposited on an 8% Tris-Glycine gel. For Aβ immunoprecipitation, the remaining secretate was incubated overnight with a 100-fold dilution of 6E10 (Eurogentec, Seraing, Belgium) and protein A agarose beads (Invitrogen, Carlsbad, CA, USA). Beads were washed twice with 1X RIPA and subjected to Tris/tricine 16.5% polyacrylamide gel. Proteins were transferred onto nitrocellulose membranes and incubated overnight with the 6E10 monoclonal antibody at a 1/1000 dilution. Immunological complexes were detected with a goat anti-mouse peroxidase-conjugated antibody (1/2000 dilution). Chemiluminescence was recorded using a Luminescence Image Analyser LAS-4000 (FujiFilm, Tokyo, Japan) and quantifications were performed using the MultiGAUGE analyser software V3.0.

### 2.5. Sandwich ELISA of Secreted and Intracellular Aβ

Cells were grown in 6-well plates and allowed to secrete Aβ as described above. Intracellular Aβ peptides were recovered as described [19]. Aβ40 and Aβ42 were measured using human Aβ40 and Aβ42 ELISA kits (Invitrogen, Carlsbad, CA, USA). The minimal detectable amount of human Aβ42 is <10 pg/mL and human Aβ40 is <6 pg/mL.

### 2.6. Western Blotting

SorLA, neomycin, APP, sAPPα, CTFs and β-tubulin were separated on Tris-glycine (8%) or Tris-tricine (16.5%) gels. Proteins were transferred onto Hybond-C membranes (GE Healthcare, Boston, MA, USA) and then probed with the following antibody: anti-SorLA (antibodies-online, Aachen, Germany), anti-neomycin (Merck Millipore, Darmstadt, Germany), 22C11 (anti-N-terminal sequence of APP), 2H3 (anti-human Aβ targeting sequence 1–12 that reacts with the N-terminus of sAPPα but not sAPPβ); BR188 (anti-C-terminal sequence of APP, provided by Dr. M. Goedert); anti-PS1-NT raised against residues 1–65 of human PS1 (kind gift from Dr. Fraser); anti-PS2-Loop raised against residues 269–394 corresponding to the from the intracellular loop region of human PS2 (provided by Dr. Thinakaran; anti-APH1aL raised against the C-terminal region of human APH1aL (kind gift from one of us, PSGH); anti-PEN2 raised against the last 24 amino acids of human PEN2 (CR8, Covance); anti-nicastrin was a goat polyclonal antibody raised against the N terminus of human nicastrin (sc-14369, Santa Cruz Biotechnology, Inc., Dallas, TX, USA); anti-β-tubulin (Sigma, St. Louis, MO, USA). Immunological complexes were revealed with either anti-mouse or anti-rabbit peroxidase antibodies (Beckman Coulter, Fullerton, CA, USA), followed by electrochemiluminescence. 

### 2.7. In Vitro γ-Secretase Assay

The γ-secretase assay by means of reconstituted membranes was carried out as described previously [20]. An equal amount of membranes preparations was incubated overnight with a recombinant C100-FLAG corresponding to the β-secretase-derived APP fragment harboring a methionine residue in position 1 [20]. Aβ and AICD-FLAG were detected by Western blotting on a 16.5% Tris/tricine gel and revealed with either anti-Aβ 2H3 or anti-FLAG antibodies, respectively.

### 2.8. BACE1 Fluorimetric Assay

BACE1 activity was followed with (7-methoxycoumarin-4-yl)-acetyl-SEVNLDAEFRK(2,4-dinitrophenyl)-RRNH2; 10μM, R&D Systems, Minneapolis, MN, USA) in absence or in the presence of β-secretase inhibitor I (50μM, PromoCell, Heidelberg, Germany) as described previously [21]. BACE1 activity corresponds to the β-secretase inhibitor-sensitive fluorescence recorded at 320 and 420 nm as excitation and emission wavelengths, respectively.

### 2.9. α Secretase Activity on Intact Cells

Plated confluent HEK293 cells were pretreated for 30 min at 37 °C with 1 mL of PBS supplemented with or without the zinc metalloprotease inhibitor *o*-phenanthroline (100 μM), then the quenched fluorimetric α-secretase substrate JMV2770 (10 μM, provided by Dr. Hernandez) was directly added to the cultured cells for various times as described [22]. At each incubation time period, medium (100 μL) was collected and fluorescence was recorded in a 96-wells plates at 320 and 420 nm as excitation and emission wavelengths, respectively. After removal of the last sample, cells were resuspended in lysis buffer (10 mM Tris/HCl, pH 7.5, 150 mM NaCl, 0.5% Triton X-100, 0.5% deoxycholate, and 5 mM EDTA); protein concentrations were determined by the Bradford method [23], and all fluorimetric values were normalized according to protein contents.

### 2.10. Neprilysin Activity Measurements

Neprilysin (NEP) activity was followed as described previously [24]. Briefly, cell homogenate samples (50 μg of proteins) were incubated in a final volume of 100μL containing NEP substrate (Suc-Ala-Ala-Phe-7AMC, 20 μM, Sigma, St. Louis, MO, USA) in the absence or presence of the NEP inhibitor phosphoramidon (10 μM, Sigma, St. Louis, MO, USA). NEP activity was considered as the phosphoramidon-sensitive fluorescence recorded at 390 and 460 nm as excitation and emission wavelengths.

### 2.11. In Vitro Cathepsin B Activity Assay

HEK293 cells were lysed mechanically in homogenization buffer (250 mM sucrose, 1 mM EDTA, 5 mM Hepes pH 7.4) using, firstly, a Dounce homogenizer, then a syringe. The cell suspension was centrifuged for 5 min at 850× *g*, then the resulting supernatant was further centrifuged for 90 min at 20,000× *g*. The pellet (membrane-enriched fraction) was resuspended in Tris-HCl (10 mM, pH 7.5) and all samples were adjusted to 6 μg/μL before analysis. Cathepsin B activity was monitored as described [19] by incubating samples (60 μg of protein extracts) in a final volume (100 μL) of acetate buffer (25 mM, pH 5.5, L-cysteine HCl, 8 mM) containing cathepsin B substrate (carboxybenzoyl-Arg-Arg-7-Amido-4-methylcoumarin (100 μM, Sigma, St. Louis, MO, USA) in the absence or presence of leupeptin (10 μM, Sigma, St. Louis, MO, USA). Specific cathepsin B activity was considered as the leupeptin-sensitive fluorescence recorded at 320 nm (excitation) and 420 nm (emission) using a fluorescence plate reader (FLUOstar Omega, BMG Labtech, Ortenberg, Germany). Fluorescence was recorded every 5 min during 150 min and cathepsin B activity was calculated as the slope in initial velocity conditions, i.e., in the linear part of the curve corresponding to the initial 30 min.

### 2.12. Statistical Analysis

Statistical analyses were performed with PRISM Software V8.0.1 (Graph-Pad Software, San Diego, CA, USA) by using the unpaired Student’s *t*-test for pairwise comparisons.

## 3. Results

Expression and fate of SorLA mutant proteins in wt-APP- and swe-APP-expressing cells.

Since SorLA protein is involved in APP lysosomal sorting [25] and because wt-APP and swe-APP traffic differently [26], we have first examined the influence of *SORL1* mutations on APP physiopathological maturation in both wt-APP- and swe-APP-expressing HEK293 cells. First, we assessed wt and mutant SorLA expressions after *SORL1* cDNA transient transfection. We show that wt, SorLA^511^ and SorLA^924^ were similarly expressed in wt- or swe-APP cells (Figure 1A, upper panel). We envisioned the possibility that the proteasome could contribute SorLA degradation. Lactacystin, a proteasome inhibitor [27] that potentiates the recovery of AD-related proteins [28,29] indeed enhances wt, SorLA^511^ and SorLA^924^ expressions in both wt- and swe-APP cells (Figure 1B).

Interestingly, as previously described [16], wt-APP expression appeared increased by wt-SorLA expression, but this augmentation was not potentiated by SorLA mutations (Figure 2A).

In order to confirm these observations, we established stable transfected CHO cells expressing either wild-type or mutated SorLA. In agreement with the above-described data, we detected clones expressing wt-SorLA, SorLA^511^ and SorLA^924^ proteins (selected clones for further analyses are indicated in red circles (Appendix A)) the expressions of which were enhanced by the proteasome inhibitor MG132 (Appendix A). 

These stably transfected cells also proved useful to examine the putative co-localization of SorLA proteins with wt-APP (see expressions of wt-APP in Appendix A). Appendix A (lower panels) indicates that wt-SorLA, SorLA^511^ and SorLA^924^ were readily detectable in these cells and colocalized with wt-APP (Appendix A, see merge panels). Closer analysis of wt-APP and wt-SorLA colocalization indicates that both are readily detectable in intracellular vesicular structures, identifiable by punctiform labeling in the cell cytoplasm (Appendix A, right panels) and that this colocalization appears to be partly disrupted when SorLA is mutated at 511 or 924 (Appendix A, left panel). This above set of data concludes that: (1) wt, SorLA^511^ and SorLA^924^ expressions are not modulated by the nature of the APP expressed (wt- or Swe-APP) nor by the cell type and transfection procedures (transient transfection in HEH293 or stable transfection in CHO); (2) wt, SorLA^511^ and SorLA^924^ expressions are potentiated by proteasome inhibitors; (3) wt-APP and SorlA co-localize at a subcellular level and 511 and 924 SorLA mutations reduce this colocalization.

### 3.1. Influence of Wild-Type SorLA and Its Mutants on Endogenous APP Expression and on Its Non-Amyloidogenic Proteolysis

Since SorLA modulates APP trafficking, we first examined whether cellular expressions of wt-APP and/or swe-APP could be affected by SorLA mutations in transiently transfected HEK293 cells. We first observed a slight but statistically significant increase in endogenous wt-APP expression triggered by wt-SorLA expression (Figure 2A). These data appear similar in stably transfected cells, which display an increased wt-APP expression linked to wt-SorLA expression (Appendix A). However, neither endogenous APP (Figure 2A), overexpressed wt-APP (Figure 3A) nor swe-APP (Figure 3B) expressions were affected by SorLA mutations in both transient (Figure 2 and Figure 3) and stable (Appendix A) transfectants.

APP undergoes both constitutive and regulated non amyloidogenic processing by α-secretase, which leads to sAPPα secretion and concomitant formation of its C-terminal counterpart C83. We show that endogenous secreted sAPPα was reduced in cells overexpressing wt-SorLA (Figure 2B). As expected, wt-APP expression increases sAPPα in stably transfected cells (compare mock and DNA3 lanes in Appendix A). Of note, the recovery of endogenous sAPPα (Figure 2B) or sAPPα in both stably (compare DNA3 and mutant lanes in Appendix A) and transiently (Figure 4A) transfected cells expressing APP were similarly reduced by wt and mutated SorLA. SorLA mutations also did not affect sAPPα recovery in swe-APP-expressing cells (Figure 4B). This set of data indicating a lack of influence of SorLA mutations on the non-amyloidogenic pathway of APP was further confirmed by direct fluorimetric measurement of α-secretase activity by recording the phenanthroline–sensitive JMV2770-hydrolyzing activity [22] on intact wt-and swe-APP cells. Thus, no difference in α-secretase activity in wt- and swe-APP plated cells was observed (Figure 4C,D).

### 3.2. Influence of Wild-Type SorLA and Its Mutants on Aβ Peptides and γ-Secretase Expression and Activity

We aimed at examining the influence of SorLA on Aβ production/recovery in our transient and stably transfected cells. We first measured total Aβ peptides recovery in secretates of wt-APP-expressing cells by immunoprecipitation (Appendix A) in the presence of the inhibitor phosphoramidon in order to prevent neprilysin-dependent degradation of Aβ peptides [24,30,31,32]. As previously reported, wt-SorLA reduces total Aβ, the recovery of which was similarly triggered by SorLA^511^ and SorLA^141^ (Appendix A). It is well known that Aβ is mainly a mix of Aβ40 and Aβ42, in which the latter accounts for about 10% of total Aβ and that slight modifications of Aβ42/40 ratio could drive toxic phenotypes in cells [33]. Since the tiny modulation of Aβ42 could have been underscored in the total Aβ immunoprecipitation procedure, we have delineated the respective levels of Aβ40 and Aβ42 by sensitive ELISA. As previously described, Aβ42 levels correspond to about 10–15% of Aβ40 (compare values in DNA condition in Appendix A). However, we did not observe SorLA-related modifications of Aβ40 and Aβ42 recoveries, except for the 924 SorLA mutant that triggers a faint increase of secreted Aβ42 (Appendix A) but not Aβ40 (Appendix A). Overall, the above data were confirmed in wt-APP-expressing transient transfectants. Thus, neither total Aβ (Figure 5A) nor Aβ40 and Aβ42 (Figure 5B) measured in secretates by ELISA were affected by SorLA mutations in APP expressing cells (Figure 5A,B). Total secreted Aβ was also not affected by SorLA mutants in swe-APP-expressing cells (Figure 5A).

Aβ peptides can also aggregate and settle intracellularly [34]. Noticeably, Aβ42 is particularly prone to aggregation and its intracellular proportion relative to total Aβ is augmented. We thus examined the levels of both intracellular Aβ40 and Aβ42 peptides in transiently transfected cells. As expected, intracellular Aβ42 levels relative to Aβ40 are drastically increased (compare black and empty bars in wt condition in Figure 5C). However, here again, wt- and mutant SorLA-expressions did not affect intracellular levels of Aβ40 and Aβ42, whatever the APP species examined (Figure 5C).

The last proteolytic step yielding Aβ peptides is accounted for by γ-secretase, a heterotetrameric complex composed of ApH1, nicastrin, Pen-2 and presenilin 1 or 2 that harbors the catalytic core [35,36,37,38,39]. Thus, we attempted to confirm the lack of effect of SorLA proteins on secreted and intracellular Aβ by examining the expression of all components of the γ-secretase complex and by measuring its catalytic activity. Figure 6A clearly shows that none of the components expression was affected by wt- of SorLA mutants, in both wt- and swe-APP-expressing cells. Finally, we have previously reported a procedure aimed at reconstituting functional γ-secretase in cell membranes and by measuring its activity by means of recombinant C100 [20]. This fragment, which corresponds to β-secretase-derived cleavage of APP (to which a N-terminal methionine has been added, see Experimental procedure), allows monitoring Aβ peptides and their corresponding C-terminal counterpart AICD productions upon γ-secretase cleavage only [20]. Figure 6B indicates that expressions of wt and mutated SorLA do not affect γ-secretase activity measured in swe-APP-expressing cells.

Overall, the above-described data obtained by multiple and complementary approaches consistently demonstrate that SorLA mutations do not affect secreted Aβ and intracellular Aβ40/42, and do not modulate γ-secretase expression and activity.

### 3.3. Influence of Wild-Type SorLA and Its Mutants on APP C-Terminal Fragments and β-Secretase Activity

The amyloidogenic pathway occurring on APP always consists of a rate-limiting catalytic step by the β-secretase BACE1 [40] that yields a C-terminal (CTF) fragment referred to as C99. C99 can undergo a α-secretase-mediated hydrolysis generating another CTF called C83 [41,42,43]. Subsequently, both C99 and C83 produce the transcription factor AICD upon γ-secretase-mediated cleavage. We examined the putative influence of wt- and mutated SorLA on CTFs expressions in wt-expressing stably transfected cells. As expected, in the control condition, wt-APP increases the levels of both C83 and C99 (compare mock and DNA3 lanes in Appendix A), but C83 expression was more abundant than C99 (Appendix A) in accordance with our previous observation that a significant part of C83 production was derived from C99 [41]. In these cells, wt- and mutated SorLA similarly reduce the expressions of C83 and C99 (Appendix A).

As expected, expressions of both CTFs were enhanced by the Swedish mutation (compare wt (−) lanes in Figure 7A,B) in agreement with previous reports on the influence of this APP mutation on β-secretase activity [44,45]. In both wt- and swe-APP-expressing cells, wt- and mutated SorLA similarly reduce the expressions of C83 and C99 (Figure 7A,B).

A previous report also indicated that CTFs expression could be enhanced by alkalinization [46], likely through a protection against proteolysis by acidic hydrolases [19,47]. This was clearly confirmed in Figure 7A,B where NH4Cl drastically enhanced both C99 and C83 expressions. This was apparently not due to a blockade of γ-secretase cleavage since NH4Cl did not affect γ-secretase components expressions in both cell lines, whatever the nature of SorLA examined (Figure 6A,B).

Both Aβ (Figure 5) and C99 (Figure 7) productions that require BACE1-mediated APP cleavage are insensitive to SorLA mutations. This consistent set of data was further strengthened by direct monitoring of BACE1 activity by fluorimetric enzymatic assay [21]. Thus, we did not detect any modulation of BACE1 by wt- or mutated SorLA in both wt-APP- and swe-APP-expressing cells (Figure 7C).

### 3.4. Influence of Wild-Type SorLA and Its Mutants on Neprilysin and Cathepsin B activities

In sporadic AD, Aβ and CTF accumulation are not due to increased production but rather to age-related defects in their catabolic processes. Several enzymes are implicated in Aβ degradation, but neprilysin appears consistently proposed as the main degrading enzyme [31,48,49]. Concerning C99 and C83, we and others have documented their destruction in lysosomal compartment by acidic hydrolases, including cathepsin B [19,47]. In this context, in order to complete our global view of both maturation and degradation processes, we assessed the putative influence of wt- and mutated SorLA on neprilysin and cathepsin B activities. Figure 8 illustrates the lack of influence of SorLA mutations on neprilysin (Figure 8A) and cathepsin B (Figure 8B), activities in both wt-APP- and swe-APP-expressing cells.

## 4. Discussion

Alzheimer’s disease is a neurodegenerative pathology with complex etiology. A subset of cases with early onset and rapid progression are due to rare autosomal dominant mutations on APP and presenilins 1 and 2, while most of cases occur lately and are of sporadic origin. However, sporadic cases can be also influenced by a genetic component since some appear linked to risk factors associated with mutations on a still growing number of genes. 

The *SORL1* gene is the only gene to date that has been proposed to contribute to both EOAD and LOAD [8]. This gene encodes a protein SorLA, which binds to APP and promotes its retrieval in the Golgi complex and its retrograde transport from endosomes to the Golgi [16]. Further, SorLA is involved in the targeting of Aβ peptides to the lysosomal compartment [15]. Overall, this suggested that *SORL1*-associated mutations could well contribute to AD pathology by modulating APP routing and proteolytic processing. However, close examination of data precluded to draw firm conclusions concerning the mechanisms that are defective in mutated SorLA-expressing cells or animal models. Particularly, there exist discrepancies between cellular and in vivo modulation of APP expression triggered by SorLA [15]. Further, functional consequences on APP processing and more precisely on Aβ40 and Aβ42 recoveries appeared highly variable according to the nature of the *SORL1* mutation and Aβ species examined [16]. This led us to examine, in depth, the putative influence of *SORL1* mutations on the physiopathological maturation of APP as well as on the fate of various APP catabolites.

Because SorLA is involved in APP trafficking [50] and because APP routing is affected by the Swedish mutations, we first carried our study on cells expressing either wt- or swe-APP. In order to bring insights on expressions and fate of SorLA mutants, we assessed the influence of the proteasome inhibitor lactacystin [51]. This pharmacological treatment was previously shown to affect APP or secretases-related proteins [29]. We observed a potentiation of SorLA^511^ and SorLA^924^ upon proteasome inhibition in both wt- and swe-APP-expressing cells. 

SorLA^511^ and SorLA^924^ expressions did not affect APP expression in both transiently or stably transfected cell models when compared to wt-SorLA. Our data envisioned the whole amount of APP because we considered that overall, this would reflect both intracellular and membrane-associated counterparts and thus, consists of a final readout of expression, traffic and clearance of APP. Our data agree with a previous study showing that *SORL1* mutants did not affect total APP expression in HEK293 cells [16]. However, it was of importance to confirm the cellular localization of wt-APP and SorLA. Our immunohistochemical analysis established first that wt-APP and wt-SorLA indeed co-localized in stably transfected CHO cells and, second, that the SorLA mutations moderately but significantly alter this co-localization. 

Although wt-SorLA reduced the levels of secreted sAPPα in both transfectant models, we did not observe SorLA variants-associated modulation of sAPPα. Previous data concerning secreted α- and β-secretases-derived APP fragments led to contrasted observations. Thus, Vardarajan and coll. reported on an enhancement of sAPPβ by common and rare SorLA variants with no change linked to wt-SorLA [16]. Caglayan et al. showed that in vivo expression of wt-SorLA did not nor alter sAPPα levels [15]. Conversely, Cuccaro et al. showed a drastic reduction of sAPPα triggered by wt-SorLA and an increase observed after expression of two SorLA mutants linked to EOAD [13]. This could not be accounted for by distinct cell models as both studies were carried out on HEK293 cells expressing swe-APP. This emphasizes the fact that the influence on α-secretase-mediated processing of APP could be differently affected by the nature of the SorLA mutations. In our study, to strengthen our observations, we directly measured α-secretase activity on plated cells. We confirm that α-secretase activity and C83 (the sAPPα C-terminal APP counterpart also referred to as α-CTF) levels were not modulated by SorLA^511^ and SorLA^924^ expressions. 

Previous studies on the influence of SorLA mutants on Aβ levels also led to contrasted conclusions. For instance, Cuccaro and colleagues reported on very faint increases of secreted Aβ42 triggered by two SorLA mutants to levels that remained lower or close to that recovered in mock-transfected cells [13]. Some of these mutants (SorLA T588I) even did not modify secreted levels of Aβ40. Whatever the mutant studied, sAPPβ levels remain similar to those recovered in mock-transfected swe-APP-expressing cells [13]. In another study, the level of secreted Aβ40 was affected by rare, but not by common, SorLA variants, while secreted Aβ42 was modulated by both variants [16]. These discrepant data could be explained by the nature of the mutation examined, but also by the fact that Aβ secreted represents only a subset of total Aβ and that it is also important to assess the levels of intracellular Aβ. Overall, our study indicates that wt-SorLA as well as SorLA^511^ and SorLA^924^ similarly reduced both intracellular and secreted Aβ40 and Aβ42 in both transiently and stably transfected cells. This conclusion was corroborated by three independent lines of results: (1) β-secretase activity was not modulated by SorLA variants; (2) γ-secretase activity and expressions of its four protein components were not changed upon SorLA variant expressions; (3) the expression of C99, which is the precursor of Aβ, was not affected by SorLA mutations. It should be noted here that some of these data are supported by previous observations obtained with other variants. Thus, Vardarajan and colleagues did not observe any modulation of PS1 (the catalytic core of γ-secretase) levels upon expressions of variants linked to both EOAD and LOAD [16]. 

Finally, we analyzed the putative influence of SorLA mutants on the events taking place downstream to APP catabolites production. Aβ is mainly processed by neprilysin, while CTFs (C99 and C83) undergo proteolysis by acidic lysosomal proteases, including cathepsin B. As would be predictable from our above-described data, SorLA mutants did not modify these neprilysin and cathepsin B activities. Thus, neither production not clearing mechanisms are affected by the SorLA mutations examined in our work.

## 5. Conclusions

Our study does not rule out the possibility that additional mutations could well influence APP physiopathological processing. However, we can conclude that our studied mutations reported to be linked to benign or severe AD cases do not trigger their putative pathogenic phenotype through modulation of APP physiopathological maturation. Again, this does not rule out the possibility that these mutations trigger pathogenic phenotypes by influencing other cellular pathways. Particularly, further study remains to be performed to assess the possible influence of these variants on Tau-related pathology or on hippocampal atrophy as has been described for a subset of *SORL1* mutants [52].

## Figures and Tables

**Figure 1 cells-12-02802-f001:**
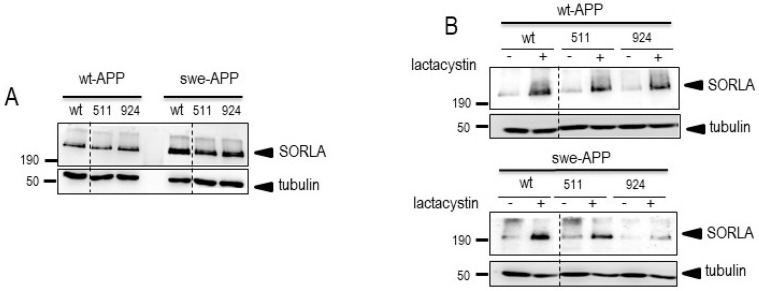
**Wild-type and mutated SorLA expressions and fate in wt-APP- and swe-APP-expressing HEK293 cells**. cDNA encoding wild-type (SorLA^wt^), SorLA^511^ or SorLA^924^ cDNA were transiently transfected in wt-APP- or swe-APP- expressing HEK293 cells (**A**) in absence (−) or in the presence (+) of lactacystin (**B**) as described in experimental procedure. Twenty-four hours after transfection, cells were harvested then SorLA and tubulin expressions were monitored by Western blot as described in the procedures. Gels correspond to one representative blot of 2 to 5 independent analyses. All full gels are shown in a Supplementary Material.

**Figure 2 cells-12-02802-f002:**
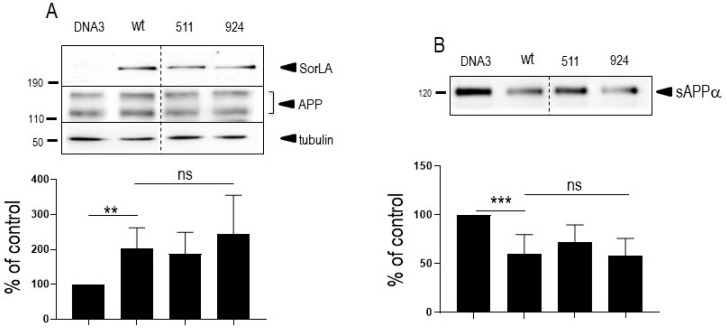
**Influence of wild-type and mutated SorLA on endogenous wt-APP expression and secreted sAPPα**. Empty pcDNA3 (DNA3), wild-type (SorLA^wt^), SorLA^511^ or SorLA^924^ cDNA were transiently transfected in wt-APP-expressing HEK293 cells (**A**). Twenty-four hours after transfection, secretates were recovered then cells were harvested and sAPPα and APP expressions were monitored by Western blot as described in experimental procedures. Bars are densitometric analyses expressed as percent of control (DNA3)-transfected cells taken as 100 and are the means ± SEM of 3 (**A**) or 4 (**B**) independent determinations. **, *p* < 0.01; ***, *p* < 0.001; ns, non-statistically significant. All full gels are shown in Appendix A.

**Figure 3 cells-12-02802-f003:**
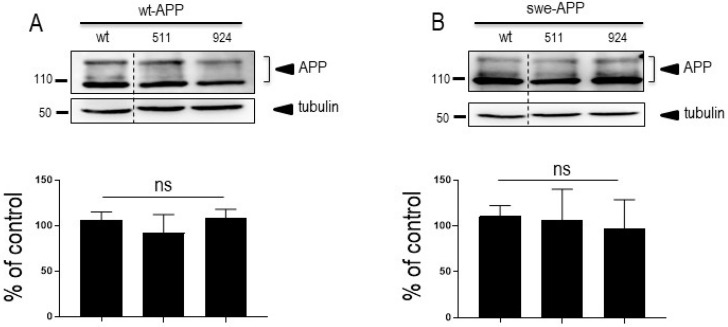
**Influence of wild-type and mutated SorLA on wt-APP and swe-APP expressions**. cDNA encoding wild-type (SorLA^wt^), SorLA^511^ or SorLA^924^ cDNA were transiently transfected in wt-APP- (**A**) or swe-APP- (**B**) expressing HEK293 cells. Twenty-four hours after transfection, cells were harvested then APP expression was monitored by Western blot as described in experimental procedures. Bars are densitometric analyses expressed as percent of control (SorLA^wt^ cDNA transfection in corresponding cells) and are the means ± SEM of 3 (**A**) or 4 (**B**) independent determinations. ns, non-statistically significant. All full gels are shown in Appendix A.

**Figure 4 cells-12-02802-f004:**
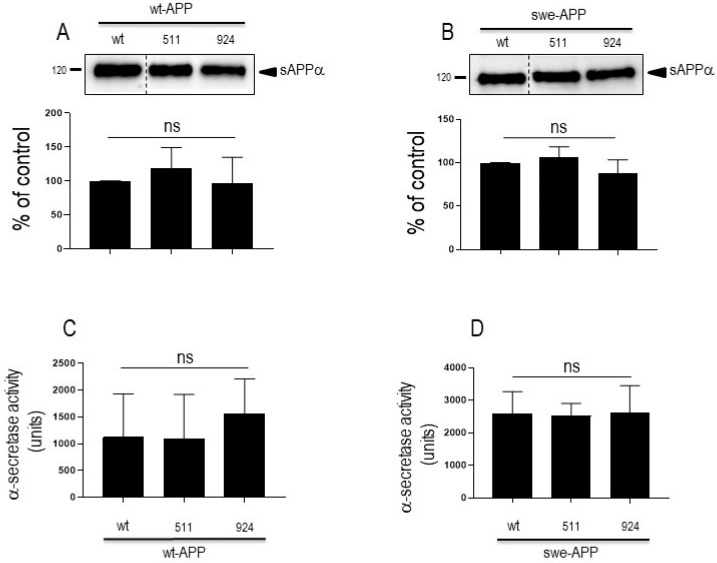
**Influence of wild-type and mutated SorLA on secreted sAPPα and α-secretase activity**. cDNA encoding wild-type (SorLA^wt^), SorLA^511^ or SorLA^924^ cDNA were transiently transfected in wt-APP- (**A**,**C**) or swe-APP- (**B**,**D**) expressing HEK293 cells. Twenty-four hours after transfection, cells were harvested then sAPPα expression was measured in secretates (**A**,**B**) by Western blotting as described in the experimental procedures. Bars correspond to densitometric analysis of sAPPα expressed as percent of control (sAPPα in SorLA^wt^ -expressing cells) and are the means ± SEM of 5 (**A**) or 4 (**B**) independent experiments. In C and D, α-secretase activity was fluorimetrically recorded on plated cells as described in the experimental procedure. Bars correspond to the phenanthroline-sensitive JMV2770-hydrolysing activity and are the means ± SEM of 9 (**C**) and 5 (**D**) independent experiments. ns, non-statistically significant.

**Figure 5 cells-12-02802-f005:**
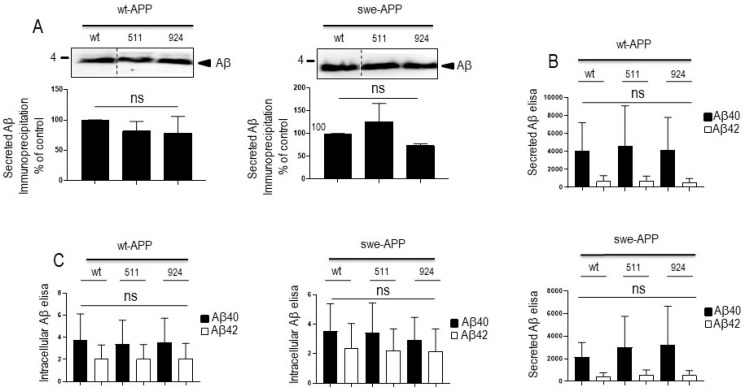
**Influence of wild-type and mutated SorLA on secreted and intracellular Aβ peptides**. cDNA encoding wild-type (SorLA^wt^), SorLA^511^ or SorLA^924^ cDNA were transiently transfected in wt-APP- or swe-APP-expressing HEK293 cells (**A**–**C**). Twenty-four hours after transfection, total Aβ (A) recovered in medium was analyzed by immunoprecipitation (**A**). Bars are densitometric analyses expressed as percent of control (densitometries in SorLA^wt^ -expressing cells) and are the means ± SEM of 3 experiments. In B and C, cells were analyzed for secreted (**B**) or intracellular (**C**) Aβ40 and Aβ42 (**C**) by ELISA as described in the experimental procedures. ELISA quantifications are the means ± SEM of 5 independent experiments. ns, non-statistically significant. All full gels are shown in Appendix A.

**Figure 6 cells-12-02802-f006:**
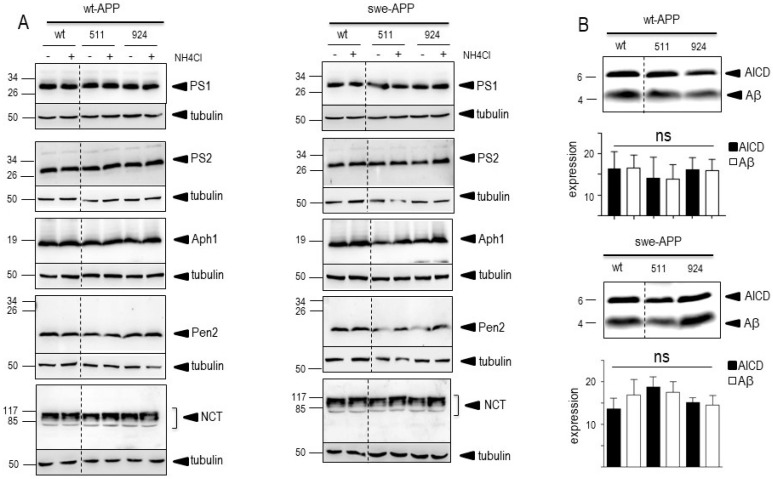
**Influence of wild-type and mutated SorLA on γ-secretase expression and activity.** cDNA encoding wild-type (SorLA^wt^), SorLA^511^ aor SorLA^924^ cDNA were transiently transfected in wt-APP- (**A**, left panel) or swe-APP- (**A**, right panel) expressing HEK293 cells, in absence (−) or in the presence (+) of NH_4_Cl. Twenty-four hours after transfection, the expressions of presenilin 1 (PS1), presenilin 2 (PS2), Aph1, Pen2, nicastrin and tubulin were analyzed by Western blotting as described in the experimental procedure. In (**B**), in vitro γ-secretase activity was measured in reconstituted membranes prepared from cells expressing SorLA^wt^ or SorLA mutants as described in the experimental procedures. Bars are densitometric analyses of Aβ and AICD expressions and are the means ± SEM of 6 independent experiments. All full gels are shown in Appendix A.

**Figure 7 cells-12-02802-f007:**
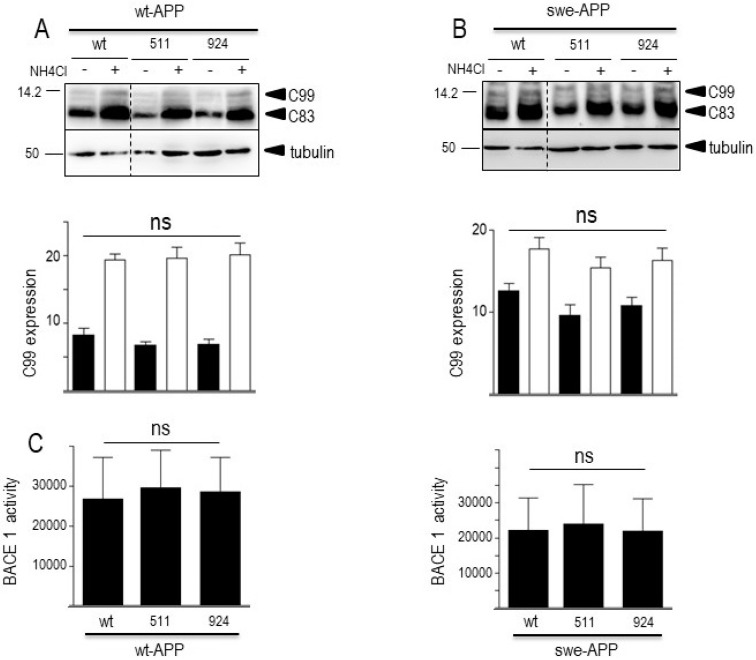
**Influence of wild-type and mutated SorLA on CTFs expression and BACE1 activity**. cDNA encoding wild-type (SorLA^wt^), SorLA^511^ or SorLA^924^ cDNA were transiently transfected in wt-APP- (**A**) or swe-APP- (**B**) expressing HEK293 cells, in absence (−) or in the presence (+) of NH_4_Cl. Twenty-four hours after transfection, the expressions of CTFs (C83 and C99) and tubulin were analyzed by Western blotting as described in the experimental procedure. Bars are densitometric analyses of C99 expression and are the means ± SEM of 7 (**A**) or 6 (**B**) independent experiments. In (**C**), BACE1 activity was fluorimetrically measured in cell homogenates as described in the experimental procedures. Bars represent the β-secretase inhibitor I-sensitive (7-methoxycoumarin-4-yl) acetyl-SEVNL-DAEFR K (2,4-dinitrophenyl)-RRNH2-hydrolyzing activity and are the means ± SEM of 5 independent experiments. ns, non-statistically significant. All full gels are shown in Appendix A.

**Figure 8 cells-12-02802-f008:**
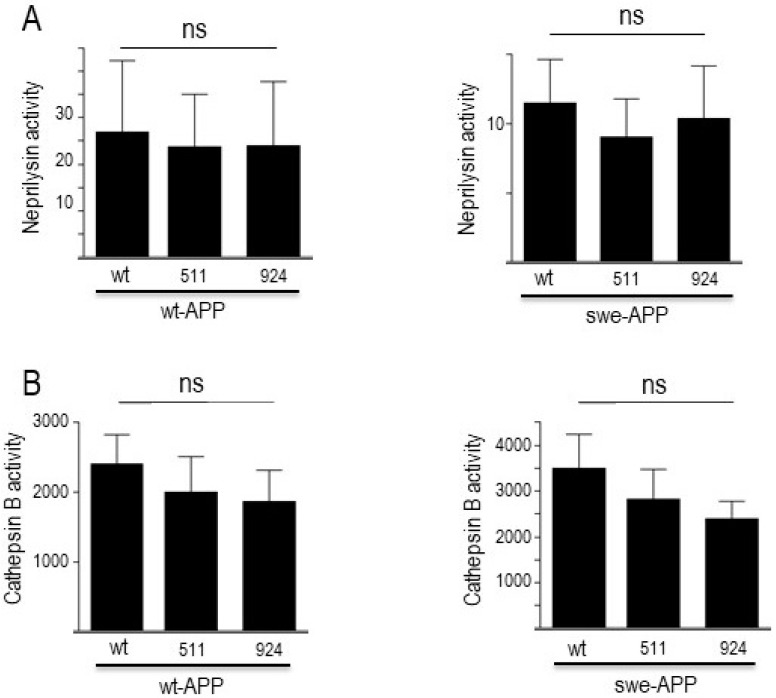
**Influence of wild-type and mutated SorLA on neprilysin and cathepsin B activities**. cDNA encoding wild-type (SorLA^wt^), SorLA^511^ or SorLA^924^ cDNA were transiently transfected in wt-APP- or swe-APP-expressing HEK293 cells. Twenty-four hours after transfection, neprilysin (**A**) and cathepsin B (**B**) activities were fluorimetrically measured with either Suc-Ala-Ala-Phe-7AMC (in the absence or presence of the NEP inhibitor phosphoramidon) or with carboxybenzoyl-Arg-Arg-7-Amido-4-methylcoumarin (without or with leupeptin) for neprilysin and cathepsin B, respectively. Bars correspond to the inhibitor-sensitive hydrolyzing activities and are the means ± SEM of 5 independent experiments. ns, non-statistically significant.

**Table 1 cells-12-02802-t001:** Point mutations in bold and corresponding mutant.

**hSORL1 Point Mutation (Nucleotide)**	**Forward Primer**	**hSorLA Point Mutation (Amino Acid)**
G1531C	5′-GGC-TCA-GTG-CGA-AAG-AAC-TTG-GCT-AGC-AA-3′	G511R
A2771G	5′-GAT-GTG-AAG-TGG-CCC-AGT-GGC-ATC-TCT-GTG-3′	N924S

## Data Availability

Data are contained within the article and Appendix A.

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
