# Peer review of "Potentially Pathogenic SORL1 Mutations Observed in Autosomal-Dominant Cases of Alzheimer’s Disease Do Not Modulate APP Physiopathological Processing"

_cells, 2023, doi:10.3390/cells12242802_

Round 1
Reviewer 1 Report
Comments and Suggestions for Authors
Over recent years, increasing evidence confirms that SORL1 is a cross-disease gene since rare SORL1 variants are found in different neurodegenerative phenotypes.
The authors have chosen three SORL1 variants (classified as likely pathogenic) associated with either benign (SorLA924) or severe (SorLA141 and SorLA511) Alzheimer’s disease phenotypes and performed a comprehensive functional analysis in two cellular models. In a cellular model of transiently transfected HEK293 cells expressing either wild-type APP or APP with Swedish mutation, the production of various APP cleavage products was assessed. The authors also analyzed expressions and activities of enzymes modulating APP processing, such as α-, β- and γ-secretases and degrading enzymes.
Comments:
1. Mutated variants SorLA141 and SorLA511 appear to disrupt colocalization with wt-APP partially. This finding deserves mentioning in the abstract.
2. Line 61-64– please provide references here.
3. A table with basic information on 3 SorLA variants would be a helpful addition to supplementary data, e.g. from the varsome database featuring their frequency, current ACMG criterium (likely pathogenic?), pathogenicity prediction, associated phenotype, refs, etc.
4. Could you comment on the apparent discrepancy regarding the SorLA141 expression pattern between your results vs. already published results (e.g. Rovelet‑Lecrux et al 2021, https://doi.org/10.1186/s40478-021-01294-4). Have you tried to detect mRNA carrying this variant with TaqMan assay?
5. Supplementary material
In suppl Fig 2 please provide a description for panels – what color corresponds to APP or SorLa
Comments on the Quality of English Language
6. English – there are minor spelling mistakes
Line 61 – benign or severe?
Line 230- Please correct HEH293
Author Response
see uploaded file

Reviewer 2 Report
Comments and Suggestions for Authors
The article is devoted to popular theme - studies of reason of AD disease. The idea of article is enough good, but is arguemented by only one link. The design of study shoud be implemented with controls for most of data presented or data should be better discussed.
In general the article leaves a bad impression because the authors confuse the protein names, figures numbers. This makes the artice is very difficult to understand, requiring you to constantly check each sentence against original research links and databases.
I attach list of comments below, it contains main inconsistencies, but not all. Authors must take manuscript editing seriously.
1. P. 2, line 52. The correct protein name - APP, in UNIPROT base and in the article Rogaeva et a., 2007. Please correct it.
2. P.2, line 53. The correct cleaved fragments names– sAPPβ and sAPPα (UNIPROT base). Please correct it.
3. Capture to figure 2. “cDNA encoding empty cDNA” doesn’t specify exactly what it meant.
4. Supplementary – didn’t indicated what are marked by red circles.
5. Mock and DNA3 – doesn’t specify exactly what it meant.
6. P.7, line 250 – Which protein expression is increased – wt APP or βAPP?
7. P.7, lines 268-271. Please rewrite this sentence “Of note…….SORLA”. It isn’t clear what you mean. In this manner it doesn’t correlate with gels.
8. P.7, line 273 – βAPP or APP? Please check the correct protein names and cells throughout the article.
9. P.9 lines 305 – Please check the names of the pictures you mention are correct. Also the results on Suppl. Fig. 4A-4C are different: the αβ level on Fig 4A is increased, but αβ42 level and αβ40 level on Fig.4B-C isn’t increased.
10. The author should to explain their position about mutant form SORLA141. They said that it isn’t expressed in sufficient amount and then analyze its influence on process. If the effect of the missing protein is the same as that of the present one (for example, wt SORLA), the entire results of paper are called into question.
11. P.12, lines 403-404. The sentence “Figure 8 illustrates…swe-APP-expressing cells” is written incorrectly. It appears that wt SORLA influence on activity of neprilysin and cathepsin B, and the mutant forms do not. As the same time, on the Figure 8 we see that SORLA doesn’t influence on the activity of their activity at all.
Author Response
see uploaded file

Round 2
Reviewer 2 Report
Comments and Suggestions for Authors
The authors are seriously corrected the manuscript and eliminated most of the comments.
Only one serious lack wasn't corrected - question about the expression of SORLA 141 mutant. The author's arguments are not convincied. Since the authors has written about puzzling reduction of CTFs (Fig 7; p. 11, lines 373-374) they must prove the presence of this mutant protein. I can propose two variant:
1. To exclude data with this mutant from the manuscript.
2. To prove (by using an Antibody from another supplier or in another way) the presence of this protein in the cells.
Author Response
We have removed all data concerning the mutant 141 of SorLA and modified the text accordingly.